# Association between Wait Time for Transthoracic Echocardiography and 28-Day Mortality in Patients with Septic Shock: A Cohort Study

**DOI:** 10.3390/jcm11144131

**Published:** 2022-07-16

**Authors:** Jiezhao Zheng, Qilin Yang, Tianyu Kong, Xiaohua Chen, Rundong Wang, Jiaxian Huo, Weichao Huang, Deliang Wen, Xuming Xiong, Zhenhui Zhang

**Affiliations:** 1Department of Critical Care, The Second Affiliated Hospital of Guangzhou Medical University, No. 250 Changgang East Road, Haizhu District, Guangzhou 510260, China; jiezhaozheng@126.com (J.Z.); yangqilin@gzhmu.edu.cn (Q.Y.); ktyfancy@126.com (T.K.); cxhmeet1024@126.com (X.C.); wrddrw1997@126.com (R.W.); huojx163@163.com (J.H.); deliangw@163.com (D.W.); 2The Second Clinical College, Guangzhou Medical University, Guangzhou 510180, China; hwc1362092086@163.com

**Keywords:** wait time, transthoracic echocardiography, septic shock, 28-day mortality, MIMIC-III

## Abstract

**Background:** the optimal timing of Transthoracic echocardiography (TTE) performance for patients with septic shock remains unexplored. **Methods:** a retrospective cohort study included patients with septic shock in the MIMIC-Ⅲ database. Risk-adjusted restricted cubic splines modeled the 28-day mortality according to time elapsed from ICU admission to receive TTE. The cut point when a smooth curve inflected was selected to define early and delayed group. We applied propensity score matching (PSM) to ensure our findings were reliable. Causal mediation analysis was used to assess the intermediate effect of fluid balance within 72 h after ICU admission. **Results:** 3264 participants were enrolled and the risk of 28-day mortality increased until the wait time was around 10 h (Early group) and then was relatively flat afterwards (Delayed group). A beneficial effect of early TTE in terms of the 28-day mortality was observed (HRs 0.73–0.78, all *p* < 0.05) in the PSM. The indirect effect brought by the fluid balance on day 2 and 3 was significant (both *p* = 0.006). **Conclusion:** early TTE performance might be associated with lower risk-adjusted 28-day mortality in patients with septic shock. Better fluid balance may have mediated this effect. A wait time within 10 h after ICU may represent a threshold defining progressively increasing risk.

## 1. Introduction

Septic shock is a severe manifestation of sepsis induced by infection, which is characterized by circulatory and cellular metabolism abnormalities [1]. Despite the improvement of treatments in recent years, the mortality caused by sepsis among the patients with critical ill still keep high and it bring a heavy medical burden. Given the higher risk of mortality of septic shock, in range of 40–50% [2], initiating resuscitation immediately is recommended to maintain the hemodynamic status stability [3].

Assessment of volume is part of hemodynamic monitoring. Optimal fluid management is one of the cornerstones of hemodynamic management in shock. Fluid responsiveness can be further indicated by pulse index continuous cardiac output (PiCCO), the rapid administration of a bolus of intravenous fluid or a passive leg-raising (PLR) test, central venous pressure (CVP) monitoring, etc. [2]. However, it was known that each methods had certain limitation, for example, CVP was affected by cardiac function, circulating blood volume and vascular tension. Echocardiography (echo), as an important complementary examination, is also proven to be one of the best bedside methods to evaluate preload, which is of great significance for the treatment and management of patients with septic shock. Hemodynamic parameters obtained through echo include cardiac tamponade, cardiac output (CO), ventricular cavity size, left ventricular systolic function, right ventricular function, Intracavitary pressures, fluid responsiveness [4]. The fluid responsiveness is reflected by vena cava diameter and respiratory variations or PLR-induced systolic volume variations.

As we know, while the noninvasive nature and quick availability of bedside echocardiography make it widespread use, using appropriately echocardiography is helpful to evaluate cardiac function and had important guiding effect on volume resuscitation, application of vasoactive drugs and other treatments, which may improve patients outcomes [5]. Bedside cardiac ultrasound has been suggested to use in ICU for critical ill patients because of their sudden changes in condition that required rapid evaluation [6]. A recent study involving patients with sepsis had demonstrated that use of transthoracic echocardiography (TTE) was associated with a lower 28-day mortality [7]. The benefit consistently persisted in patients during septic shock [8]. However, there are few studies focusing on the association between wait time for completing echocardiography and mortality in patients with septic shock. Thus, we investigated the relationship and hypothesized that early TTE performance might bring better 28-day outcomes by influencing fluid management in patients with septic shock. 

## 2. Methods

### 2.1. Data Source

All the data in present study was extracted from Medical Information Mart for Intensive Care (MIMIC)-III (version 1.4)—a real-world and freely-available clinical database which comprised more than 60,000 patients who stayed in intensive care unit (ICU) of Beth Israel Deaconess Medical Center (BIDMC, a large tertiary care hospital in Boston, MA, USA) between 2001 and 2012 [9]. The MIMIC-III database was populated with data that had been acquired during routine hospital care, mainly including: archives from critical care information systems, hospital electronic health record databases and Social Security Administration Death Master File. The database had been widely used in academic research and was fully recognized. We were approved to use the database after completion of the “Protecting Human Research Participants” course. One author Jiezhao Zheng had obtained certification which numbered 10007310. Our study was reported according to the Strengthening the Reporting of Observational Studies in Epidemiology (STROBE) guidelines [10].

### 2.2. Ethical Approval

The Institutional Review Boards (IRBs) of the Massachusetts Institute of Technology (Cambridge, MA, USA) and Beth Israel Deaconess Medical Center (Boston, MA, USA) approved the use of the MIMIC-III database for the present study (certification number: 10007310). Since the database does not contain protected health information and this study was retrospective in nature, a waiver for the requirement for informed consent was included in the IRB approval.

### 2.3. Study Population and Setting

We enrolled patients with septic shock which defined as sepsis patients administered with norepinephrine within 24 h after ICU admission [1]. In addition, Sepsis was diagnosed based on Angus criteria [11]. Nonadults (age > 18 years) and patients who stayed less than 24 h in the ICU were excluded. The patients who had performed transthoracic echocardiography before ICU admission were not enrolled. For patients admitted to the ICU more than once, only the first ICU stay was considered.

### 2.4. Main Exposure

The primary study variable was wait time for TTE which defined as the total time elapsed from ICU admission until TTE performed (in hours). The time of the first TTE performance was exacted from the text information in table named “noteevents”.

### 2.5. Covariates

Our study’s covariates were selected based on previous study in which shown that they were associated with the mortality in sepsis patients [12]. We included the following variables: demographic characteristics, vital signs, laboratory tests, interventions [using of mechanical ventilation, sedative drugs and renal replacement therapy during the first 24 h of ICU admission] and comorbidities [congestive heart failure (CHF), atrial fibrillation (AFIB), chronic renal disease, liver disease, chronic obstructive pulmonary disease (COPD), coronary artery disease (CAD), stroke, and malignant tumor]. All the comorbidities were identified on ICD-9 codes as previous reported [7]. The severity of the diseases was estimated by Sequential Organ Failure Assessment (SOFA) scores, Simplified Acute Physiology Score (SAPS) II, Acute Physiology Score (APS) III, and Oxford Acute Severity of Illness Score (OASIS). The first values of vital signs and laboratory tests on the first day were taken.

### 2.6. Outcomes

The primary outcome was 28-day mortality. A 28-day observation window is the most common follow-up period measured in literature [13,14]. Secondary outcomes included 90-day mortality, the number of days with weaning from mechanical ventilation, transferring out of ICU and without vasopressors within 28 days after ICU admission, and fluid balance within 72 h after ICU admission.

### 2.7. Missing Data

Variables missing more than 20% were removed from this analysis. The missing values were replaced by the mean or median values for continuous variables with missing values less than 5% [15].

### 2.8. Statistical Analysis

A risk-adjusted restricted cubic splines with 4 knots [16] for the association of the time elapsed from ICU admission to the first TTE performance carried out with 28-day mortality was performed. Any nonlinear relationship between wait time for TTE and 28-day mortality could be assessed using spline regression. Rather than arbitrarily dividing patients into early and delayed TTE groups, the association between delayed TTE and mortality was graphically represented to visualize an inflection point (in hours), if one existed. Further examination of the inflection point used two-piece-wise Cox regression model. We applied a recurrence method to determine the inflection point where the smoothing curve started to change and became eminent. The inflection point was moved along a pre-defined interval and detected the inflection point that gave the maximum model likelihood [17]. Inflection point was selected as the cut point to dichotomize wait time for TTE as early or delayed group.

We generate a propensity score matching (PSM) among those early TTE group and delay TTE group using a multivariable logistic regression model. A 1:1 nearest neighbor matching algorithm was applied using a caliper width of 0.01. A standardized mean difference (SMD) was used to examine the PSM degree. The SMD and statistical significance of the observed differences were then calculated with the paired t test for continuous covariates and chi-square test for categorical covariates. Furthermore, the variables mentioned above as covariates were selected to generate the propensity score. The estimated propensity scores were used as weights. Pairwise algorithmic (PA) [18], standardized mortality ratio weight (SMRW) [19] models were used to generate a weighted cohort to adjust the baseline confounders, thus reflecting more truly the independent association between wait time and mortality. E-value was used to assess the effect of unmeasured confounding on study results [20]. 

Subgroup analyses according to age, comorbidities and relevant clinical interventions were performed. We also conduced sensitivity analyses after separating patients with no TTE from delayed group to verify the stability of our results.

Causal mediation analysis (CMA) [21] is a method to identify intermediate variables (or mediators) that lie in the causal pathway between the treatment and the outcome. In view of the results of echocardiograph often affected the fluid administration, we set early TTE performance as the treatment and fluid balance within 72 h after ICU admission as mediator variables to explore whether the effect of early TTE performance on the primary outcome is mediated by the fluid balance in our study.

A descriptive analysis was performed for all participants. Categorical variables were expressed as a number of percentages (%) and compared using the chi-square tests. Continuous variables were expressed as mean and standard deviation (SD) when normally distributed or median and interquartile range (IQR) when skewed. The One-Way ANOVA, Kruskal-Wallis test were used for comparison, as appropriate. For survival analysis, Kaplan-Meier curves were depicted and compared by log-rank test.

All analyses were performed using the statistical software packages R. version 3.4.3 (R Foundation for Statistical Computing, Vienna, Austria) and Free Statistics software versions 1.4. A two-tailed test was performed and *p* < 0.05 was considered statistically significant.

## 3. Results

We identified 17,420 sepsis patients according to the Angus definition and a total of 3264 patients with septic shock who met inclusion were included in the study, with 2220 (68%) had completed TTE (Flowchart in Appendix A). The mean (SD) age of all patients was 67.8 (16.0) years and 53.9% were male. The median (IQR) wait time for TTE was 20.6 (10.3–60.2) hours. We further used restricted cubic splines to flexibly model and visualize the relation of wait time and 28-day mortality (Figure 1). The risk of 28-day mortality started to increase 5% per hour until the wait time was around 10 h and then was relatively flat afterwards (*p* for non-linearity = 0.015, Appendix A). Early group (544 patients, 16.67%) was therefore defined as TTE performed within 10 h of ICU admission. The rest of the participants (2720 patients, 83.33%) included patients who waited for TTE more than 10 h or did not receive after ICU admission was classified as the delayed group.

The patients in early group had lower severity scores except SOFA score which was not found statistical difference. In addition, they received more ventilation (85.7% vs. 72.9%) and sedative management (80.7% vs. 66.1%) compared with delayed group. With the use of propensity-score matching, 532 pairs patients were matched and most of covariates were balance between two groups. All the baseline variables were comparable (Appendix A).

### 3.1. Primary and Secondary Outcomes

Early group had a significantly lower 28-day mortality (20.4% vs. 29.3%, *p* < 0.001, Figure 2). The risks of death within 28 days in the early and delayed group were visualized by K-M curve and compared, which supported this result. (Appendix A). (HR = 0.65, 95%CI 0.54–0.80, *p* < 0.001, Figure 2). After adjusted for all covariates, multivariate regression analysis demonstrated a significant beneficial effect of early TTE in terms of the 28-day mortality and the adjusted hazard ratio was 0.74 (95%CI 0.60–0.91, *p* = 0.004, Figure 2). Furthermore, the association still remained stable in multivariate analysis using PSM adjusted for propensity score, PA, SMRW. The HRs were 0.73–0.78, all *p* < 0.05 (Figure 2). The E-value was 1.77.

The potential outcomes that might attribute to the benefits of early TTE were summarized in Table 1. Similarly, lower 90-day mortality was found in early group after PSM (27.4% vs. 35.9%, *p* = 0.004). Not only was the number of days free from mechanical ventilation and vasopressor in the early group significantly longer, but they were also earlier transferred from ICU within 28 days. Compared with the delayed group, although the fluid balance to the early group was lower during the first three days after ICU admission, statistic difference was only found on day 2 and 3.

### 3.2. Causal Mediation Analysis

CMA was applied by us to explore the direct and indirect effects of early TTE performance on 28-day mortality. The indirect effect was significant when the volume of fluid balance on day 2 and 3 were mediator variables. The total effect were 0.048 (95%CI 0.008–0.085; *p* = 0.016) and 0.056 (95%CI 0.013–0.098; *p* = 0.012), the ADE were 0.041 (95%CI 0.002–0.078; *p* = 0.040) and 0.046 (95%CI 0.004–0.090; *p* = 0.026), the ACME were 0.007 (95%CI 0.002–0.011; *p* = 0.006) and 0.009 (95%CI 0.003–0.017; *p* = 0.006), the proportion of the effect mediated were 14.8% (95%CI 3%–62.4%; *p* = 0.022) and 17.0% (95%CI 3.9%–69.2%; *p* = 0.018), respectively (Figure 3). However, there was no significant indirect effect when the volume of fluid balance on day 1 acted as mediator (ACME −0.001; 95% CI −0.002–0.001; *p* = 0.57). Therefore, we deemed the beneficial effect of early TTE on 28-day mortality was partly mediated through the volume of fluid balance within 3 day after ICU admission (except day 1).

### 3.3. Subgroup Analyses and Sensitivity Analyses

In the subgroup analyses, for patients who were at advanced age or with coexisting conditions (diabetes, respiratory disease, renal disease), the associations between early TTE performance and lower 28-day mortality were statistically significant, and no significant interaction was detected. Patients with mechanical ventilation or indwelled arteriovenous catheters appeared to have a stronger association than those without, and there were no interactions (Figure 4).

To exclude the effect of patients who did not undergo TTE on the delayed group results, we performed a sensitivity analysis after dividing the delayed group into the late group (wait time longer than 10 h) and the no TTE group. In addition, we found the curve in Figure 1 showed a significant downward trend after the wait time approached 40 h, so we further divided the late group into patients with waiting time between 10 h and 40 h and patients with waiting time over 40 h. After compared with the early group, rise likelihood of 28-day mortality in patients with waiting time between 10 h and 40 h and in the no TTE group existed (HR 1.26, 95%CI 1.01–1.58, *p* = 0.043; HR 1.89, 95%CI 1.51–2.38, *p* < 0.0001; respectively, Appendix A). However, there was a reduced tendency of 28-day mortality in patients with waiting time over 40 h, but it did not have statistic difference in the comparison (HR 0.99, 95%CI 0.77–1.26, *p* = 0.915).

## 4. Discussion

Our retrospective study revealed that TTE performed as soon as possible within 10 h after admission was associated with lower 28-day mortality in patients with septic shock. After adjusting for potential clinical factors using a variety of statistical methods, this association was reliable. To our knowledge, this is the first study to analyze time as a continuous variable in hours and identify a time-to-TTE threshold associated with increased risk of poor prognosis among adults undergoing septic shock after ICU admission. Moreover, Fluid balance may be a part of the intermediate factor that was influenced by TTE and thus impact on the outcome, despite TTE was a diagnostic and hemodynamic assessment technique rather than a treatment.

Bedside ultrasound, incorporated as the complement of the traditional physical examination [22], had been introduced as a pocket-sized device in intensive care and was widely used over the past decade, making it one of the most powerful diagnostic and therapeutic tools available to critical care practitioners [23]. Echocardiography had been recommended for some conditions with rapid progression in critical ill patients in order to determination of cardiopulmonary instability and evaluation of effective volume state [24]. Similarly, sepsis, or even septic shock, has the same characteristics and needs early fluid resuscitation for improvements of prognosis.

It has been proven that in many diseases, patients who undergo ultrasound have better outcomes than those who do not. For example, Young et al. suggested that delays to echo (˃4 days) would increase the risk of complications including shock in patients with suspected infective endocarditis [25]. For patients with shock, the results of laboratory tests were not available in a timely manner for the initiation of therapy in the acute situations, echo could be considered to be rapidly applied [26]. Previous studies conducted by Feng et al. [7]. and Lan et al. [8]. shown that TTE was associated with a 28-day mortality benefit in a population of sepsis or septic shock, which was consistent with the higher risk of mortality in patients without TTE compared to the early group in the sensitivity analysis of this study. However, the optimal time in which to initiate a performance of TTE is still unclear in sepsis shock. In the present study, we proposed to perfect TTE as soon as possible within 10 h and it was expected to be considered as an effective supplement to 6-h bundles [27].

To the best of our knowledge, anyone can suffer from sepsis, but those at particular risk of developing septic shock include: the older, people with pre-existing medical conditions such as diabetes, lung disease, cancer and kidney disease, and patients with catheters or mechanical ventilation [28,29]. Our subgroup analyses suggested that TTE should be performed as early as possible in most of the high-risk population indicated above. Moreover, with the increase of invasive medical procedures, the incidence of sepsis is increasing year by year. For example, the implantation of pacemaker will lead to endocarditis, and septic shock may progress to the later stage of the disease. Echocardiography is beneficial to the early diagnosis of pacemaker lead endocarditis and is of great significance to the prevention of disease progression [30]. Regretfully, due to the limited amount of data on these patients in this database, it had not been analyzed and studied, and this issue needed to be paid attention to by more prospective studies in the future.

We used CMA and found that the beneficial effect of early TTE on 28-day outcome in patients with septic shock was partly due to fluid administration. On the next two days after the first day of ICU admission, fluid balance, respectively, contributed14.8% and 17.0% to improved outcomes in our study. Therefore, we reasonably suspected that early ultrasonography may have altered the diagnosis or treatment. In a related study conducted by Kavi Haji et al. suggested that management changed in 65% of TTE participants in ICU [31]. Although our results partially interpreted the causal relationship between TTE, fluid management, and short-term mortality in patients with septic shock, it is difficult to be convinced that an isolated TTE examination would have significant influence on patient mortality [32]. Hence, it still needed to be further demonstrated in prospective studies with high quality of evidence.

Interestingly, a phenomenon was observed that in the correlation curve between wait time and risk of death within 28 days, when time was greater than 40 h, the risk of death showed a downward trend. We considered this happened because of the immortal time bias. Immortal time was defined as a span of cohort follow-up during which the outcome under study could not occur because of sufficiently long survival time prior to exposure [33]. We realized that the median length of an ICU stay was 2.1 days and Septicemia was the disease with the highest mortality rate which was up to 48.9% among the adult patients in MIMIC-III database [34]. Given the short ICU stay time and high mortality rate, we could infer that when the wait time for TTE exceeds 40 h, the condition of these patients might have improved, and whether TTE was performed might have little impact on the outcome. We conducted a sensitivity analysis using a time-dependent Cox regression model to avoid immortal time bias [35] and our results remained stable. Although we initially included the patients with no TTE into the delayed group, which compared with the early group, this might cause certain influence on the results. However, we independently isolated patients with no TTE from our study cohort in the sensitivity analysis showing that the early group was still significantly associated with better outcomes compared with the patients with wait time longer than 10 h or no TTE.

There were several noteworthy limitations in our study. First, this was a retrospective cohort study in which the potential for residual confounding might exist and some important clinical variables were missing due to the limitation of integrity of data collection in the database. We had tried our best to control the confounders and used E-value to further estimate the effect of unmeasured confounding on the results. After various statistical evaluations and adjustments, our results were relatively stable. Furthermore, since our study only involved a single medical center with a large scale, it could be assumed that the conditions for patients to obtain medical equipment resources (such as ICU staffing and echo acquisition) were the equivalent. In this way, data acquisition and the results of this study were more reliable. Secondly, the definition of sepsis in our study was diagnosed according to Angus criteria, which was different from Sepsis 3.0, which limited the generalizability. In addition, it was difficult to determine the exact time of sepsis diagnosis and the reasons for the TTE because of the deficiencies in retrospective study based on database data. Thirdly, we only considered the influence of single TTE on prognosis, and did not evaluate the influence of multiple TTE examinations and their frequency on prognosis. Even so, TTE was recommended in the guideline as a dynamic measure to guide reliable fluid resuscitation, especially after initial resuscitation [36]. At the same time, the results of our study suggested that early TTE examination (≤10 h) was associated with better outcomes, which could be better interpreted and accepted. In addition, lactate [37] and CVP [38] have been recommended to guide fluid resuscitation in patients with septic shock, but they still have some limitations in fluid evaluation. TTE, as a complement, combine with these two factors to comprehensively assess volume status to guide resuscitation may be beneficial to improve prognosis, which warrants further investigation. Finally, the causal relationship between early TTE and 28-day mortality was not explored clearly, although we examined the mediation effects of fluid balance, these effects need to be further verified in future studies.

## 5. Conclusions

Early TTE performance might be associated with lower risk-adjusted 28-day mortality in patients with septic shock. Better fluid balance may have mediated this effect. A wait time within 10 h after ICU may represent a threshold defining progressively increasing risk.

## Figures and Tables

**Figure 1 jcm-11-04131-f001:**
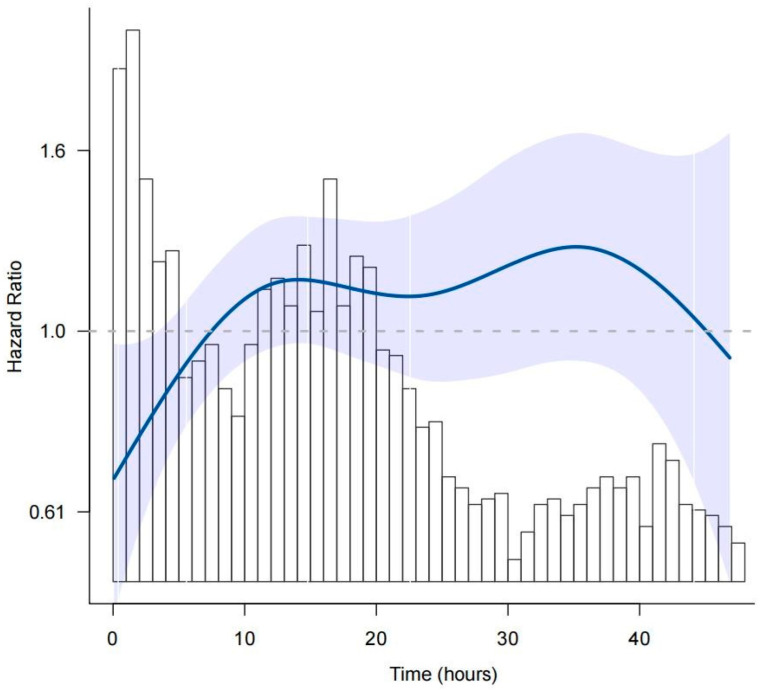
Probability of 28-day mortality according to wait time for TTE. Date were fit by a multivariable logistic regression model based on restricted cubic splines.

**Figure 2 jcm-11-04131-f002:**
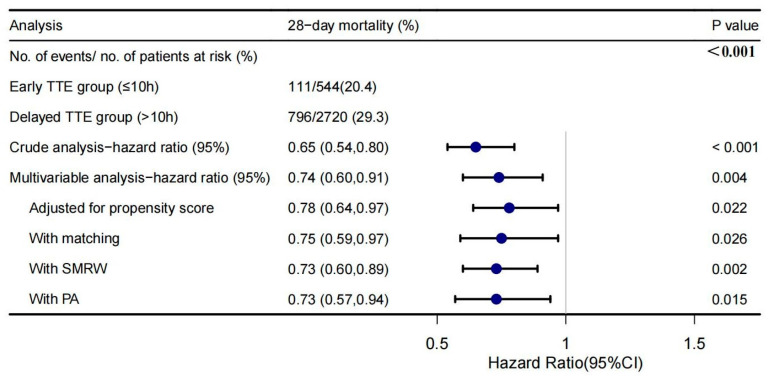
Primary outcomes of patients.

**Figure 3 jcm-11-04131-f003:**
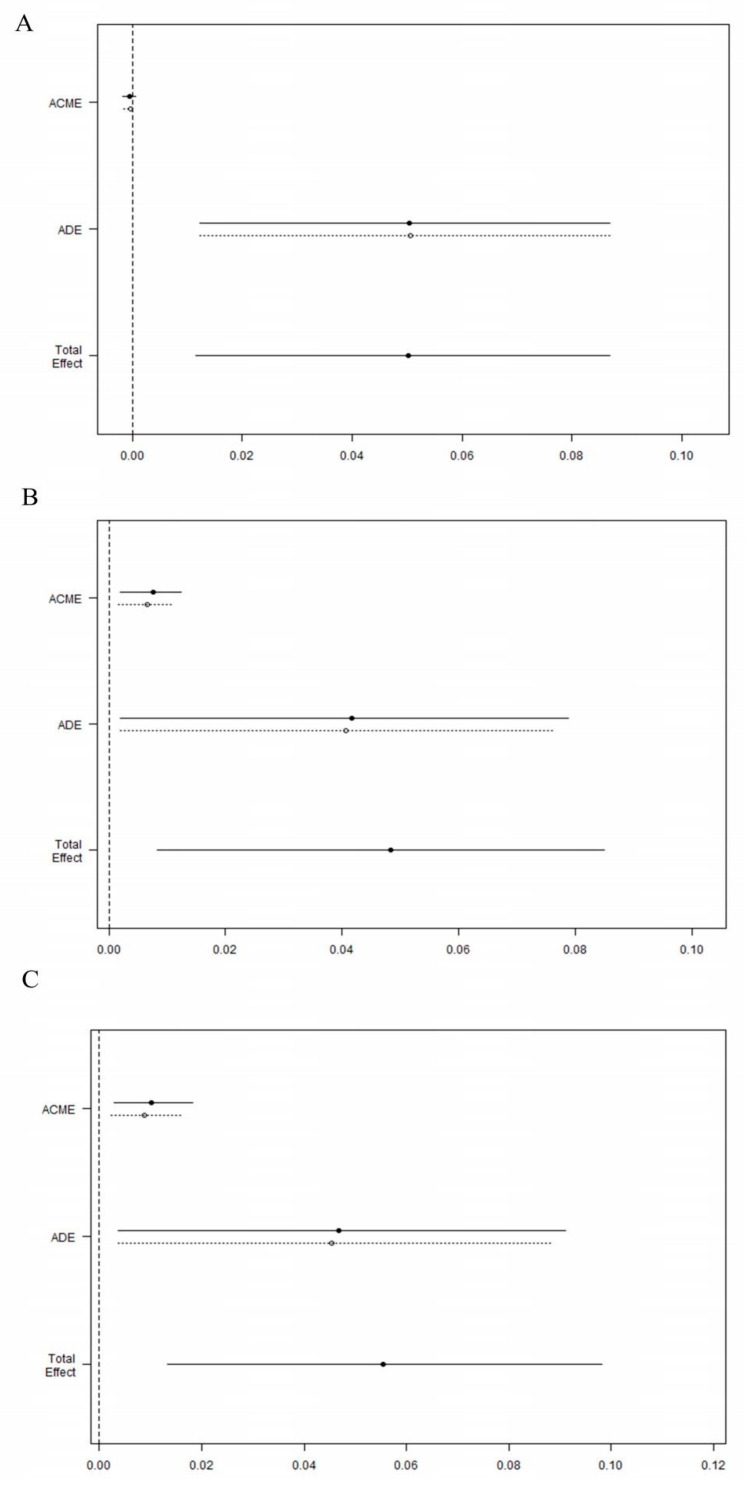
Causal mediation analysis for fluid balance on day 1 (**A**), 2 (**B**) and 3 (**C**). The solid line represents the early TTE, and the dashed line represents the delayed TTE.

**Figure 4 jcm-11-04131-f004:**
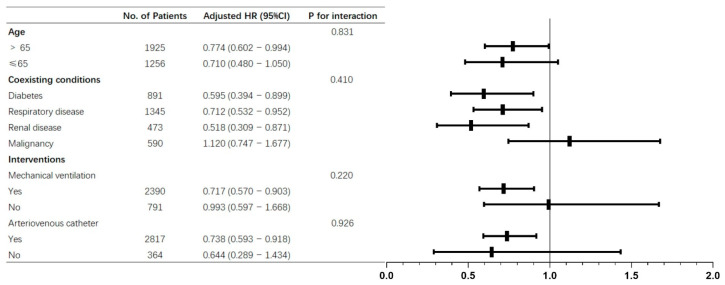
Subgroup analyses of the association between early TTE and 28-day mortality in the original cohort.

**Table 1 jcm-11-04131-t001:** Secondary outcome analysis after matching.

Secondary Outcomes	Time of TTE		*p* Value
	Early (≤10 h)	Delayed (>10 h)	
	(N = 532)	(N = 532)	
**90-day mortality, n (%)**	146 (27.4)	191 (35.9)	0.004
**The number of days in 28 days, Median (IQR)**			
Ventilation-free	22.5 (1.9–27.2)	19.8 (0.0–26.5)	0.007
vasopressor-free	25.6 (17.4–27.3)	25.0 (0.0–27.2)	0.043
ICU-free	18.1 (0.0–24.1)	14.9 (0.0–23.0)	0.008
**Fluid management (L), Median (IQR)**			
Fluid input day 1	4.4 (2.5–7.1)	4.8 (2.8–7.7)	0.080
Fluid input day 2	1.9 (1.0–3.9)	2.5 (1.4–4.3)	<0.001
Fluid input day 3	1.4 (0.5–2.6)	1.7 (0.9–3.2)	<0.001
Fluid output day 1	1.4 (0.8–2.2)	1.5 (0.9–2.4)	0.468
Fluid output day 2	1.4 (0.8–2.2)	1.4 (0.8–2.2)	0.763
Fluid output day 3	1.7 (0.9–2.6)	1.7 (1.0–2.7)	0.648
Fluid balance day 1	2.8 (1.0–5.4)	3.0 (1.0–5.9)	0.157
Fluid balance day 2	0.4 (−0.8–2.2)	1.0 (−0.3–2.9)	0.002
Fluid balance day 3	−0.2 (−1.3–1.2)	0.1 (−0.9–1.7)	0.004

## Data Availability

Data in the article can be obtained from MIMIC-III database (https://mimic.physionet.org/).

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
