# Peer review of "Association between Wait Time for Transthoracic Echocardiography and 28-Day Mortality in Patients with Septic Shock: A Cohort Study"

_jcm, 2022, doi:10.3390/jcm11144131_

Round 1

Reviewer 1 Report

This study examined the waiting time for echocardiography in patients with septic shock and concluded that testing within 10 hours of admission improves prognosis. While the focus of this study is interesting and useful in clinical practice, I would like to highlight the following key concerns that need to be addressed to improve the quality of this article.

1.       Figure 1 shows that the prognosis seems to improve after 30 hours. What are your thoughts on this?

2.       This study states the importance of TTE for fluid volume management, but is POCUS sufficient? If so, isn't central venous pressure monitoring sufficient?

3.       What parameters are important for echocardiography in patients with septic shock, and how can the results obtained be used in patient management?

Author Response

Dear Editor and Reviewers,
On behalf of my co-authors, we greatly appreciate the careful review and comments from both you and the reviewers.
We present here point-to-point responses for each of the comments of reviewers and have revised our manuscript accordingly. Revised sections are identified with red text in the paper. And we hope the revised manuscript could be acceptable for you.
Please see the attachment.
Thank you and best regards.
Yours Sincerely,
Zhenhui Zhang
E-mail: zhzhhicu@126.com
Department of Critical Care, the Second Affiliated Hospital of Guangzhou Medical University, No. 250 Changgang East Road, Haizhu District, Guangzhou, Guangdong, China

Reviewer 2 Report

The authors deal with the role of transthoracic echocardiography (TTE) in the management of septic shock by focusing on the association between wait time for completing 
 echocardiography and mortality in patients with septic shock. 

This is a retrospective study with data extracted from a data base (MIMIC-III database). MIMIC-III (‘Medical Information Mart for Intensive Care’) is a large, single-center database comprising information relating to patients admitted to critical care units at a large tertiary care hospital.

A certain criticism has to be manifested.

1) A more details has to be added regarding this  data base (MIMIC-III).

2) It is not clear if the patients have already got a diagnosis of septic shock before admission to the  CCU. In that case TTE should confirm and eventually refine the clinical assessment and most of all assess the volume status.

4) With the exclusion of cardiogenic shock all other forms of shock (so including septic shock) will require volume expansion. This can be achieved quickly through elevation  of the legs (reverse Trendelenburg) and the administration of repeated small volumes  200-300 ml over 15 to 30min of an IV fluid that will remain at least transiently in vascular space such as NS and Ringer’s lactate. And eventually norepinephrine and other vasopressor should also be added.  Then the volume status has to be reassessed with purely clinical parameters: the internal jugular top meniscus after infusion should increase at  2- 3 cm H2O in the neck above the sternal angle and concomitantly with normalization of HR , blood pressure and tissue perfusion parameters. So in my view the role of wait time for completing 
 echocardiography could make sense if in comparison with a clinical approach as above described.

So It’s important to have a head to head comparison of a seasoned clinical approach (jugular etc)  and TTE .

5) Did TTE refine the diagnosis beyond volume status assessment?

6) Sepsis can affect anyone, but those at particular risk include:

  • The very old (older than 65 years old) or very young or pregnant women
  • People with pre-existing infections or medical conditions such as diabetes, lung disease, cancer and kidney disease
  • People with weakened immune systems
  • Patients who are in the hospital
  • People with severe injuries, such as large burns or wounds
  • Patients with catheters (IVs, urinary catheters) or a breathing tube

It could be crucial to understand which risk category these septic patients belong to and that if TTE could have a different role in these  different clinical categories.  

 7) the patients with lead pacemaker endocarditis are a category on the rise. Recent reports have indicated the diagnostic role of intracardiac echocardiography in assessing lead vegetations in these pts  [1] and could be very interesting to understand how many of these pts get complicated with sepsis shock and if this shock complication could be reduced by a precocious  TTE evaluation.  That has to be added.

8) The data should be checked by a professional statistician.

Minor points

.    What is “PSM”? there is no legend in the abstract. 

.    62  “ Patients with critical ill often face a significant change..” (English) 

REFERENCES

1.         Polat G, Ugan RA, Cadirci E, Halici Z. Sepsis and Septic Shock: Current Treatment Strategies and New Approaches. Eurasian J Med. 2017;49(1):53-8.

2.         Leibovici L, Drucker M, Konigsberger H, Samra Z, Harrari S, Ashkenazi S, et al. Septic Shock in Bacteremic Patients: Risk Factors, Features and Prognosis. Scandinavian Journal of Infectious Diseases. 1997;29(1):71-5.

3.         Caiati C, Pollice P, Lepera ME, Favale S. Pacemaker Lead Endocarditis Investigated with Intracardiac Echocardiography: Factors Modulating the Size of Vegetations and Larger Vegetation Embolic Risk during Lead Extraction. Antibiotics (Basel, Switzerland). 2019;8(4).

Author Response

(The authors gave the same response as above.)

Reviewer 3 Report

The authors have analysed in this retrospective study the statistical difference between patients that performed transthoracic echography (TTE) 10 hours after Intensive Care Unit (ICU) and patients that did not performed it or 10 hours after admission in ICU. The study is incomplete and so I recommend major revisions. 

1) There is no table describing characteristics of cohort. No comparison of characteristics between the groups (age, sex...). Please, integrate this data.

2) In supplementary table 2, statistical differences are observed between the two groups for all demographic and clinical parameters. How do the authors explain that TTE is the only cause in reducing mortality to 28 days?

3) Why is Exclude patients who had all of their TTEs ordered before ICU admission (583 exclude) if performing TTE early is a predictor of mortality?

4) Several studies have analyzed the statistical difference between the groups that performed TTE and those who did not (PMID: 34353053, 29806057, 35433965). Patients who did not have TTE were added to patients who had the procedure beyond 10 hours. It would be more correct to stratify the groups by the time when the TTE was performed to understand at what time to carry out the diagnostic procedure.

Author Response

(The authors gave the same response as above.)

Round 2

Reviewer 1 Report

The submitted manuscript was well revised. There are no further comments. Thank you for your revision.

Reviewer 2 Report

The paper has been improved just a little.